# Characterizing polarization in online vaccine discourse—A large-scale study

**Bjarke Mønsted**[1], **Sune Lehmann**[1,2]*

**1** Department of Applied Mathematics and Computer Science, Technical University of Denmark, Lyngby, Denmark, **2** Center for Social Data Science, University of Copenhagen, Copenhagen, Denmark

* sljo@dtu.dk

**Data Availability Statement:** Data cannot be shared publicly, as this was a condition for IRB approval. For inquiries regarding data access, contact compute@compute.dtu.dk.

## Abstract

Vaccine hesitancy is currently recognized by the WHO as a major threat to global health. Recently, especially during the COVID-19 pandemic, there has been a growing interest in the role of social media in the propagation of false information and fringe narratives regarding vaccination. Using a sample of approximately 60 billion tweets, we conduct a large-scale analysis of the vaccine discourse on Twitter. We use methods from deep learning and transfer learning to estimate the vaccine sentiments expressed in tweets, then categorize individual-level user attitude towards vaccines. Drawing on an interaction graph representing mutual interactions between users, we analyze the interplay between vaccine stances, interaction network, and the information sources shared by users in vaccine-related contexts. We find that strongly anti-vaccine users frequently share content from sources of a commercial nature; typically sources which sell alternative health products for profit. An interesting aspect of this finding is that concerns regarding commercial conflicts of interests are often cited as one of the major factors in vaccine hesitancy. Further, we show that the debate is highly polarized, in the sense that users with similar stances on vaccination interact preferentially with one another. Extending this insight, we provide evidence of an epistemic echo chamber effect, where users are exposed to highly dissimilar sources of vaccine information, depending the vaccination stance of their contacts. Our findings highlight the importance of understanding and addressing vaccine mis- and dis-information in the context in which they are disseminated in social networks.

## Introduction

Vaccine hesitancy, defined as the reluctance or refusal to vaccinate [1], is a growing threat to global health, and is believed to be driven mainly by the 'three C's': Confidence, Complacency, and Convenience [2]. Social media platforms may potentially influence vaccine hesitancy through the former two, for example by enabling easy and wide-spread sharing of content that exaggerates the risks of vaccination and/or understating the risk of vaccine-preventable diseases [3]. Vaccine hesitancy exist on a continuous spectrum [4], where the extreme positions of rejecting or accepting all vaccines tends to be overrepresented in online settings [5].

**Funding:** This study was funded entirely by the Danish Council for Independent Research (Project: Microdynamics of Social Interactions, grant number 4184-00556a). The funders had no role in study design, data collection and analysis, decision to publish, or preparation of the manuscript.

**Competing interests:** The authors have declared that no competing interests exist.

While vaccine hesitancy is a nuanced and context-dependent phenomenon, some general factors influencing hesitancy have been identified in the literature [6]. Key among those factors are the availability of information regarding vaccines [4, 7], the accuracy of beliefs about the risks and benefits of vaccines and vaccine-preventable diseases [8, 9], social norms regarding vaccination, i.e. whether or not vaccinating is perceived as a 'normal' thing to do [5, 10], and trust in health authorities and/or the pharmaceutical industry, particularly concerns regarding commercial conflicts of interest [4, 7, 8]. The list above is by no means exhaustive, rather a few central factors which are well-described in the literature and of particular relevance for this paper.

These factors are strongly connected to the topic of social networks and online misinformation: Anti-vaccine messages on Twitter typically aim to alter the reader's perception of risks and benefits regarding vaccination, often drawing on conspiracy theories [9]. In addition to an inaccurate risk picture, anti-vaccine content on Twitter, especially during the COVID-19 pandemic, has focused on commercial interests in the pharmaceutical sector [11], and often rely on conspiracy theories in doing so [12].

The detrimental effects of reduced vaccine uptakes on public health are well described in the literature [13–18]. Somewhat paradoxically, vaccination rates have declined in part due to the success of vaccines in preventing disease, leading to complacency [19, 20]. However, online misinformation has also been linked to decreasing vaccine uptake [21–24], and outbreaks of vaccine-preventable diseases have been observed in areas where anti-vaccine activists have organized disinformation campaigns [25]. In this sense, the growing amount of online misinformation [26, 27] can be characterized as a threat to public health [28].

Countering medical misinformation in online systems is no easy task. While the scientific literature is rife with evidence which disproves the narratives outlined above [29, 30], individuals at the 'rejection' extreme of the vaccine hesitancy continuum often have a strong sense of identity regarding their stance on vaccines [31]. Individuals tend to reinterpret or disregard information if it conflicts with a stance that they strongly identify [32], an effect which has been demonstrated in numerous contexts [33] including vaccination [34].

The challenges of countering misinformation is compounded by the fact that strongly anti-vaccine individuals often form tightly knit communities in large social networks, such as Twitter [35, 36] and Facebook [37]. In such environments, evidence challenging in-group beliefs is dismissed as untrustworthy [8], and often ends up only reinforcing said beliefs [37, 38].

Therefore, study of the interplay between vaccination attitudes and vaccine-related online (mis)information is essential to inform policy [39–41], also at the community level [42].

We utilize two large datasets to study this interplay. The first (Dataset 1) is a large, random sample consisting of of approximately 60 billion tweets. The second (Dataset 2) consists of 6.75 million tweets obtained via Twitter's search API for tweets containing vaccination-related terms. Both datasets are discussed in detail in the Methods section. Using these datasets, we construct a large network which captures interactions on Twitter, and use machine learning methods to identify Twitter profiles with vaccine stances at the 'rejection' and 'acceptance' extremes of the hesitancy continuum, known colloquially as anti- and pro-vaxxers, respectively.

Based on the data and methods outlined above (which are elaborated upon in the materials and methods section), the remainder of the paper presents a number of analyses on the interplay between strong vaccination stances, social network structure, and online information.

## Anti- and pro-vaccine profiles distinct types of URLs

We use a deep neural network to classify vaccine sentiments expressed in tweets from dataset 2, and identify 'antivaxx' and 'provaxx' profiles which consistently express highly negative and positive attitudes toward vaccination, respectively. Full details regarding the stance detection

methods are presented in the materials and methods section. In the following, we assess the degree to which anti- and provaxx users tend to rely on distinct types of outside sources, what characterizes these sources, and whether interactions occur disproportionally between profiles with similar stances. After estimating vaccine sentiment in the individual tweets, we estimate the vaccine *stance* of each individual profile. We define a profile's stance as provaxx or antivaxx if at least half of vaccine-related tweets posted by the profile are assigned a probability of at least 50% of expressing pro- and anti-vaccine sentiment, respectively. Approximately 48% of profiles were assigned pro- and antivaxx stances, as some profiles had posted only a few tweets regarding vaccination, or did not unambiguously express the same attitudes toward vaccination. In terms of the vaccine hesitancy continuum [4], these labels correspond loosely to the extremes, which tend to be over represented in online discussions [5]. Note that the correspondence is not exact, because hesitance is defined in terms of *accepting* or *rejecting* vaccines, whereas the labels used here refer to *attitudes* regarding human vaccination. However, intention to vaccinate is strongly influenced by attitudes regarding vaccination [43].

The distributions of tweet sentiments and profiles stances are shown in Fig 1. Of the vaccine-related tweets in English, approximately 35% could be identified as originating from the United States. An analysis at the state level of the vaccine sentiments expressed in tweets is provided in S2 Appendix. Details on data gathering, classification, and geolocation are provided in the materials and methods section.

Of those tweets, approximately 2.65 million contain external links (URLs outside of Twitter). Many such URLs start with a *base URL*, such as youtube.com, followed by a part which specifies which subpage (e.g. which particular video) the link points to, as well as various API calls, etc. We extracted the base URL for each such link, resulting in around 100 thousand

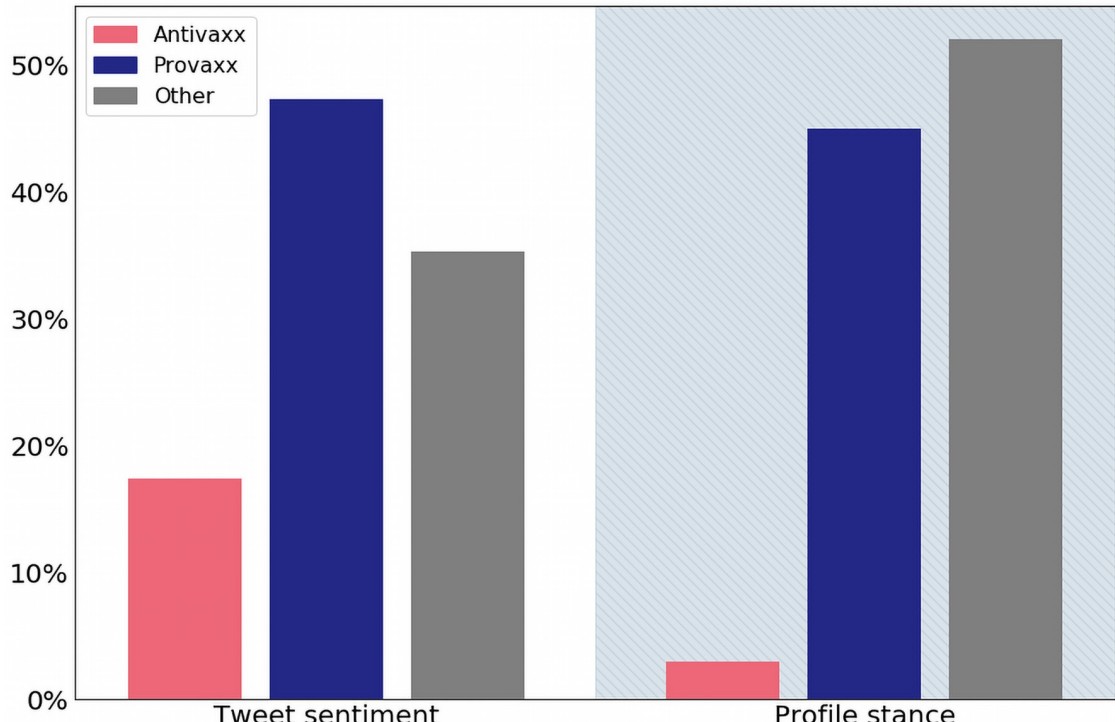

**Fig 1. Distribution of tweet sentiment and profile stance.** Tweets expressing anti-vaccine sentiment constitute an estimated 17% of vaccination-related tweets, where only about 3% of profiles stance are classified as antivaxx. Error bars are too small to depict visually, see S1 Appendix for uncertainty analyses.

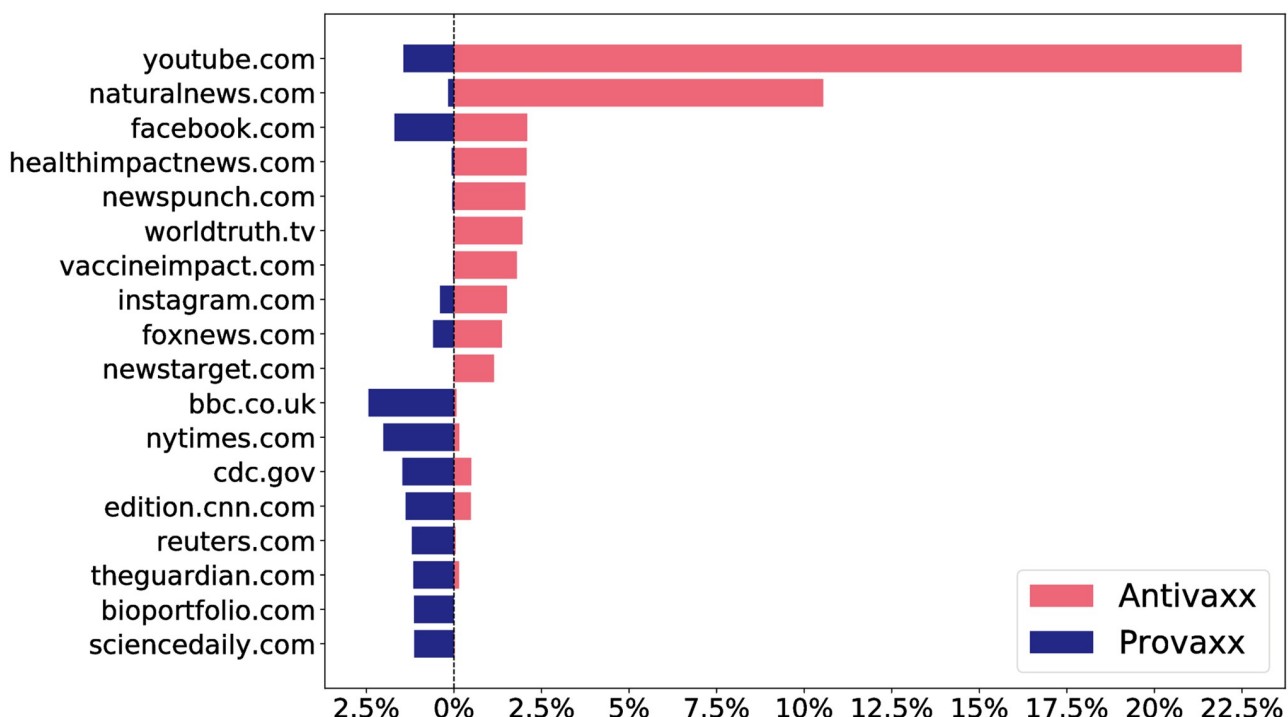

**Fig 2. The top 10 most linked to domains by strongly antivaxx and provaxx profiles.** Bar length shows percentage of the total number of links shared by profiles in the given category and hence do not sum to 100. For each domain, the red bars going right represent antivaxxers and blue bars going left provaxxers. Antivaxxers rely heavily on links to YouTube, and the page 'natural news', which promulgates pseudoscience and sells products related to health and nutrition. Provaxxers link to a wide array of news and science sites, which is why a lower overall percentage of their links are contained in the top 10. Error bars are too small to depict visually, see S1 Appendix for uncertainty analyses.

distinct base URLs. Of those, we identified the 10 most frequently used by anti- and provax-xers, resulting in 18 URLs due to two domains (facebook.com and youtube.com) appearing among the ten most common URLs for both groups. The lists of top ten URLs contained 47% of links posted by antivaxx users, and 15% of links posted by provaxx users. Comparing the most frequently used base URLs with sentiment results reveals that profiles with different stances share highly dissimilar content, as shown in Fig 2. In the top ten URLs, profiles with a pro-vaccine stance typically share content from mainstream news sites, medical or technol-ogy/science sites, and various social media sites, whereas anti-vaccine profiles tend to share content from YouTube, social media sites, and a number of sites specializing in alternative health products, pseudoscience, and conspiracy theories. For details on the categorization of links, see the Materials and methods section. The absolute number of links posted to each such domain varies a lot over time for some domains, yet the relative frequency of posts for each stance is relatively constant; visualizations and statistics are provided in S3 Appendix.

Expanding upon the above, we assign to each of the popular base URLs one or more of the following labels: "news", "social", "science", "conspiracy", "pseudoscience", "commercial". By 'commercial' we here mean sites which sell products related to (alternative) health, and so have a direct financial interest in the vaccination discourse. Exact definitions of all labels are provided in the methods section. Fig 3 shows that fraction of links posted by profiles of differ-ent stances which belong to each such category. The results summarized in Fig 3 are generally robust to changes in the sentiment threshold for stance attribution, with one exception: When increasingly strict thresholds are applied, i.e. when we consider only users who share very

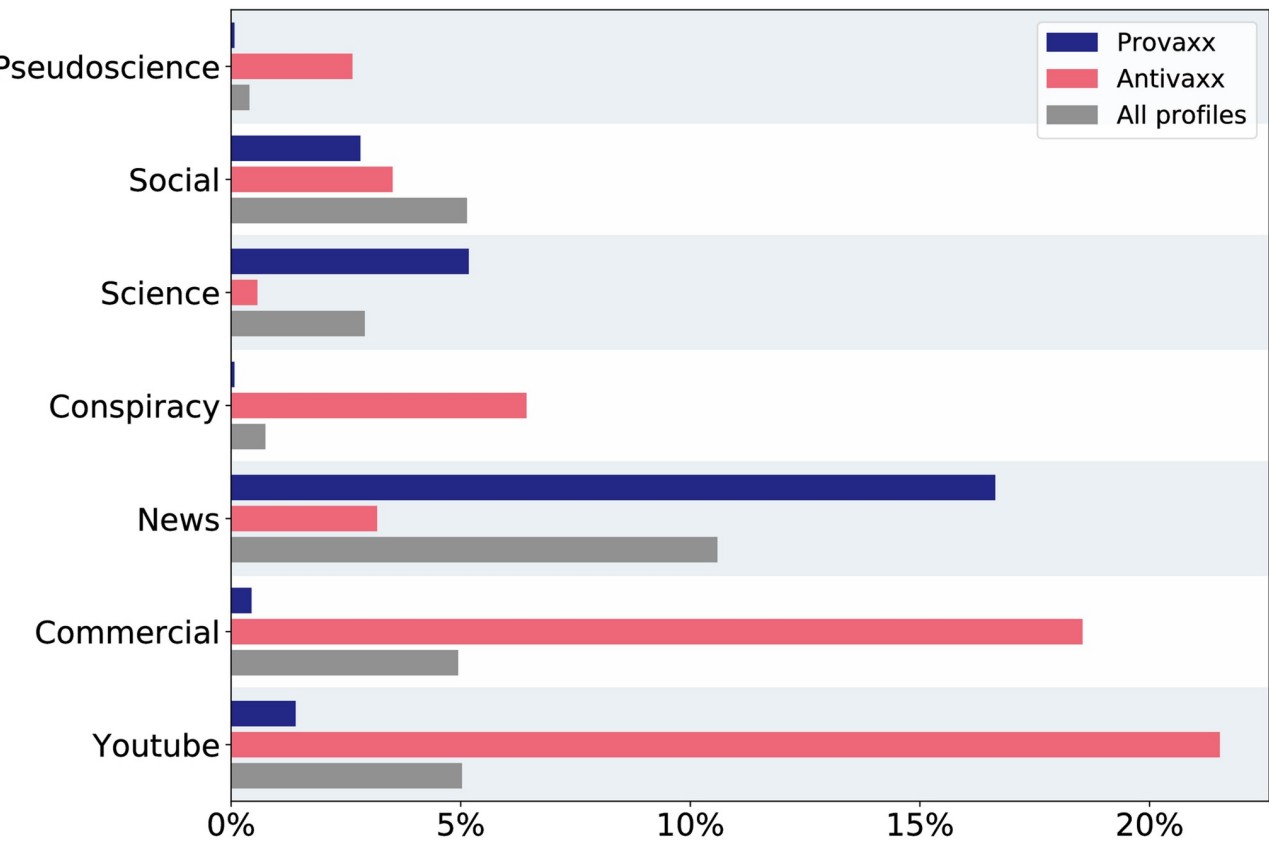

**Fig 3. Frequency of various categories of links for profiles grouped by vaccination stance (provaxx, antivaxx, or neutral).** Antivaxx profiles often post links to Youtube videos, and to sites that sell health related products and thus have a vested financial interest in the vaccine discourse. Error bars are too small to depict visually, see S1 Appendix for uncertainty analyses.

strong vaccine sentiments, the "news" category becomes more frequently linked to among anti-vaccine profiles. This is due to strongly anti-vaccine profiles disproportionally posting URLs which point to Fox News. Details and figures regarding that analysis are presented in S1 Appendix.

The most frequently occurring link category for pro- and anti-vaxxers are news sites, and Youtube links, respectively. The second most frequent category among antivaxx profiles is commercial sites profiting from selling health related products. This finding is unexpected, as common reason for vaccine hesitancy is mistrust in medical research due to perceived financial conflicts of interest and industry ties to pharmaceutical companies [44]. Links to pseudoscience and conspiracy sites are also posted disproportionally by profiles with a strong anti-vaccination stance.

## Polarization and epistemic echo chambers

Using Dataset 1, we construct a large network representing observed mutual interactions between profiles on Twitter. In this network, profiles are linked if there exists a reciprocal @-mention, or a reciprocal retweet, within a 3-month time window. We refer to the interaction network as the MMR (mutual mention/retweet) network for this reason. Additional details regarding the MMR network, as well as some analyses of the network structure and temporal stability, are provided in the materials and methods section. Links are constructed in

this fashion for consecutive time windows, where the number of such 3-month time windows in which two users have interacted can then be viewed as the weight of the link. In addition to thresholding on this weight, which is illustrated in Fig 5, the graph may be thresholded according to the number of vaccine-related tweets from each user, such that only nodes corresponding to users who posted at least a desired number of tweets with vaccination-related keywords are retained in the graph.

We initially consider a version of the MMR graph constructed using very strict criteria for node and link inclusion, then subsequently investigate the effects of easing those criteria. We first include, in each time window, only nodes that are assigned a pro- or anti-vaccine stance. Further, we only include links between nodes that interacted in several windows. The strictness of these criteria retains only nodes which consistently express strong vaccine-sentiments, in interact repeatedly nodes that do so as well. As a consequence of the strict criteria, the resulting graph contains only 4894 nodes, of which 3359 (69%) nodes form a giant connected component. The remaining connected components are loosely scattered, have fewer than 30 nodes in each, and contain only 5.6% anti-vaccine users. 395 nodes (11.76%) in the giant connected component represent antivaxx profiles. A representation of the graph using a force layout algorithm [45] is shown in Fig 4. The interplay between the stances of users and their neighborhoods, as well as user connectivity and activity, is visualized in S4 Appendix.

The graph is heavily stratified with regards to vaccination stance. The assortativity coefficient (Pearson correlation between stance and connectedness in the graph) is $r = 0.813$.

The analyses above, however, dependend on discretely partitioning users into two distinct categories. Considering instead user stance as a continuous variable—given by e.g. the average anti-/pro-vaccine sentiment expressed in their tweets—we obtain similar findings. Discarding users with fewer than 5 vaccination related tweets, we rebuilt the interaction graph while

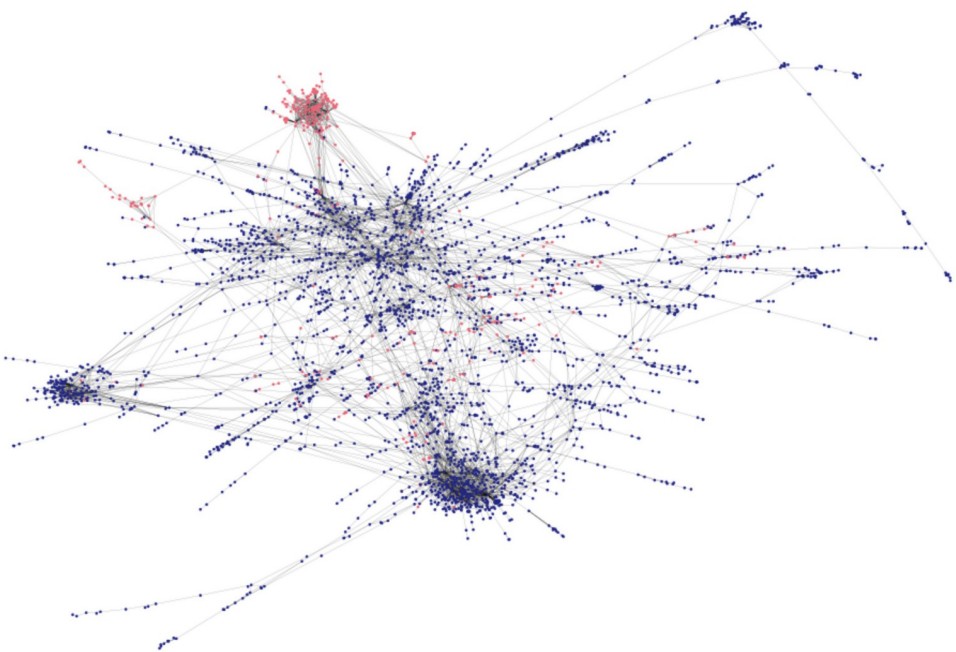

**Fig 4. Representation of the repeated mutual interaction graph from 2013–2016.** Profiles frequently interact with others who share their own stance, and antivaxx profiles are localized in relatively few, tightly nit clusters. Profiles with and anti- and provaccine stances are illustrated in red and blue, respectively. Only the giant conected component of the interaction graph is depicted.

varying the minimum number of 3-month time windows in which users must have interacted before being connected in the graph. Results on the interplay of (continuous) stance and (repeated) connectivity are summarized in Fig 5.

Correlations between the mean vaccine sentiment expressed in neighboring users' tweets were roughly the same independently of how frequently the users interacted, as shown in Fig 5a, although the number of users in the interaction graph decreases quickly when using strict inclusion criteria (additional analyses in the methods section). Similarly, we observe an anti-correlation between pro- and anti-vaxx probabilities of neighbors, which seems to diminish somewhat when considering repeated interactions. However, this decrease appears to be driven by a few nodes which have many connections, yet do not frequently discuss vaccines, as shown in Fig 5b.

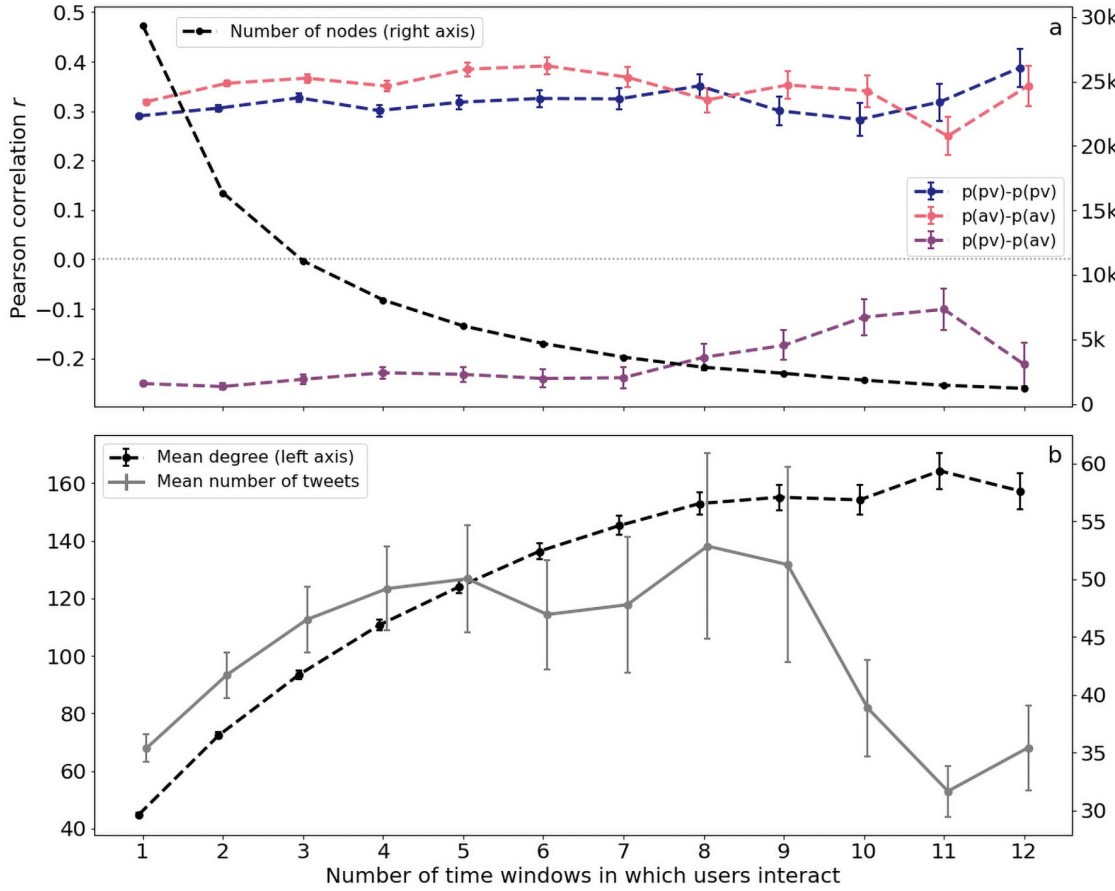

**Fig 5. Interplay between average vaccination sentiment and user interactions. a**: Users tend to disproportionally interact with users of similar stance, both in cases where users only interact during a single, and multiple, three-month time windows. Specifically, we compute for all users the average probabilities of that user's tweets expressing pro/anti-vaccine sentiment. Comparing these averages for all nodes and their neighbors, we find a positive correlation between the average pro- and antivaccine sentiments. Similarly, the average pro-vaccine sentiment of nodes exhibits a negative correlation with the anti-vaccine sentiments of their neighbors. The number of nodes in the interaction network decreases exponentially as the minimum number of time windows is increased. The negative correlation between pro- and antivaxx probabilities of neighbors tends slightly toward zero as the threshold for repeated interaction grows. **b**: As increasingly repeated interactions are considered, users in the interactions graph are increasingly well connected. However, the number of vaccination-related tweets posted by users decreases for interactions occurring very frequently, indicating that at this point, the graph likely includes users who are highly active on Twitter, yet do not discuss vaccination-related topics very often. Error bars on Pearson correlations represent one standard deviation of the Fisher-transformed variables $z$, i.e. the bounds on the error bar on a correlation $r$ of $n$ data points, is given by $\tanh(z \pm \sigma_z)$, where $z = \mathrm{arctanh}(r)$ and $\sigma_z = 1/\sqrt{n-3}$.

The finding that users interact disproportionally with other users sharing their stance, aligns with previous findings that long time anti-vaccine users of social media tend to form tightly knit clusters which exhibit a high degree of in-group solidarity [46], and in which misinformation may thrive unquestioned [47]. To qualify the latter, we turn again to the URLs most frequently shared by users discussing vaccines, shown in Fig 2, we probe regions in the MMR network around individuals of various stances and assess whether the URLs shared in those regions differ more or less from a normal distribution depending on stance.

Considering only the approximately 32 thousand users with at least 5 vaccination-related tweets, we group users based on the mean probability of antivaxx ($p_{av}$) of their tweets. We computed the deciles of $p_{av}$ for all tweets and grouped users based on which deciles their mean score fell between, i.e. one bin for mean $p_{av}$ values below the first decile, one for values between the first and second deciles, and so forth. For each such group of users, we observed the regions surrounding them in the MMR network, and extracted the URLs shared by all users who were located in that region, and who had shared at least 5 URLs and posted at least 5 vaccine-related tweets. We then computed the frequency for each of the top URLs for the regions (locally), and observed the difference from the overall (global) frequency distribution. The frequency distributions may be interpreted as maximum likelihood estimates of the probability distributions over links shared in the regions around specific users, and globally. Therefore, we quantify the difference between such distributions using the Jensen-Shannon (JS) distance [48]—an information-theoretical measure of distance between probability distributions which take values in the range between zero (no overlap between distributions) and one (identical distributions).

Fig 6 shows the JS-distances between overall link frequencies, and links shared by users adjacent to users with a given mean $p_{av}$. The figure shows that Twitter profiles that engage in online vaccine discourse are not only disproportionately connected to other users who share their stance, but that users with stronger anti-vaccine stances are also exposed to increasingly atypical sources of information. This is indicative of 'epistemic echo chambers' in online vaccine discourse in the sense that users, depending on their stance, are exposed not only to a skewed distribution of stances from other users (i.e. network homophily), but also to information sources that are highly dissimilar to those typically partaking in the overall discussion. Although we do not attempt to explain how these echo chambers arise in the first place, we can point to some mechanisms described in the literature which are consistent with our results. First, it is a well-known result in sociology and network science that links tend to form between nodes that share similar attributes [49, 50]. Second, some studies indicate that people are highly selective in sharing information that aligns well with their convictions [51], which in term can cause polarization by opinion reinforcement [52], and by users cutting ties to avoid exposure to information causing cognitive dissonance [53].

## Discussion

In summary, our findings paint a picture of the vaccine discourse on Twitter as highly polarized, where users who express similar sentiments regarding vaccinations are more likely to interact with one another, and tend to share contents from similar sources. Focusing on users whose vaccination stances are the positive and negative extremes of the spectrum, we observe relatively disjoint 'epistemic echo chambers' which imply that members of the two groups of users rarely interact, and in which users experience highly dissimilar 'information landscapes' depending on their stance. Finally, we find that strongly anti-vaccine users much more frequently share information from actors with a vested commercial interest in promoting medical misinformation.

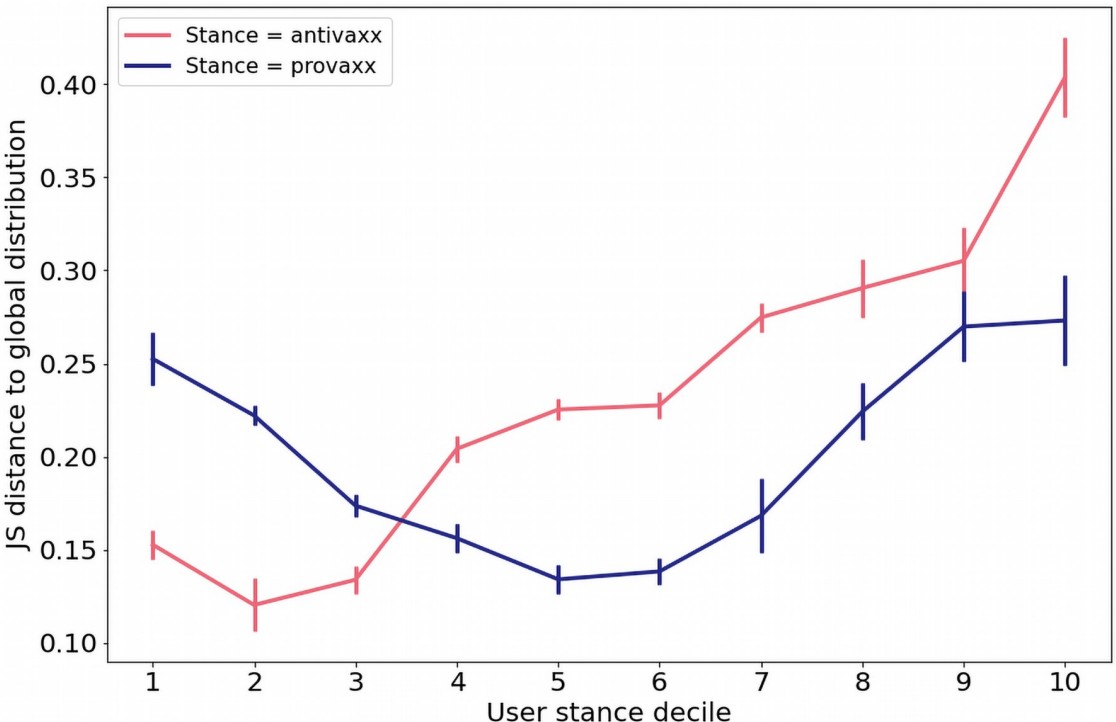

**Fig 6. Profiles that express fringe vaccine sentiments are also exposed via their interaction networks to sources of information that are highly dissimilar to link frequencies in the overall discussion.** Here we consider users who posted a minimum of 5 tweets containing vaccine-related keywords, and partition them into deciles based on their tweets' mean probability of expressing anti- and pro-vaccine sentiment. For each such decile and vaccine stance, the plot shows the Jensen-Shannon distance between the frequencies at which links from the domains shown in Fig 2 are shared in the vicinity of users in that decile, and in the interaction network overall. The error bars are computed using a bootstrap technique in which users in the target stance-decile combination where randomly sampled with replacement and the JS-distance to the overall distribution calculated. The error bars depict the standard deviations of each 1000 such samples.

One implication of these findings is that online (medical) misinformation may present an even greater problem than previously thought, because beliefs and behaviors in tightly knit, internally homogeneous communities are more resilient [37, 54], and provide fertile ground for fringe narratives [55, 56], while mainstream information is attenuated [57]. Furthermore, such polarization of communities may become self-perpetuating, because individuals avoid those not sharing their views [58], or because exposure to mainstream information might further entrench fringe viewpoints [59].

A further problem exacerbated by the structure of the debate is that, parents often base their vaccination decisions on their impression of what other parents do [60], so vaccine hesitant parents who encounter a strongly anti-vaccine community might get the impression that not vaccinating is the norm and opt not to. This risk is compounded by the fact that anti-vaccine communities are highly effective at reaching out to undecided individuals [61], which highlights the need to reach undecided individuals with accurate information to overcome vaccine hesitancy [62].

In summary, the characteristics of the online vaccine discourse may contribute to increasing vaccine hesitancy, possibly into the extreme of vaccine denial. A brief discussion of measures that have proven successful in decreasing hesitancy and increasing vaccine uptake therefore seems in order. One such measure is encouraging direct communication between hesitant individuals and healthcare professionals. Parents who interact with health care

professionals are significantly more likely to vaccinate their children [63, 64], whereas parents of underimmunized children are significantly more likely to obtain medical information online [23]. Another measure is implementing policies which incentivize vaccination or discourages rejection [65–67].

In terms of digital interventions, our findings highlight the need for measures based—not just on whether content is true or false—but on a more nuanced understanding of the interplay between vaccination attitudes, social network structure, and information sources, including actors with a vested interest in promoting false beliefs. With disinformation campaigns aiming to erode consensus [24, 68], fact-checking at the level of individual stories being shared online might need to be complemented by an understanding of the complex interplay between community structure and information content.

Future work based on the findings presented here could investigate e.g. the text content of the communication between users with highly similar and dissimilar stances regarding vaccination, as well as interactions between text topics and community structure.

## Materials and methods

This section provides details of the data analyzed and the methods employed.

### Twitter data

The work presented here relies on a large collection of data from Twitter. For clarity, we describe below two subsets of the data used in the analysis. The reason for this is that one part of the data comes from a large collection of data not specific to vaccination but well suited, due to its size, to analysis overall interactions on Twitter, whereas another was obtained by querying for vaccination-related keywords, and thus is better suited for analyses specific to vaccination. All data were collected through Twitter's public search API—no terms of service agreements were violated in collecting the data.

**Dataset 1** consists of a large collection (approximately 60 billion) of tweets [69], collected as a random 10% in 2013–2016. These data were used to a general 'interaction network' in which nodes represent Twitter profiles, and connections between nodes represent cases where both profiles either mention or retweet each other. We refer to this as the Mutual Replies/Retweets (MMR) network.

**Dataset 2** was constructed using Python to paginate backwards through the official search API for tweets containing various keywords pertaining to vaccination. A full list of the keywords queried for is: "unvaccinated", "unvaccined", "vaccinate", "vaccinated", "vaccinating", "vaccination", "vaccinations", "vaccinator", "vaccinators", "vaccine", "vaccined", "vaccinering", "vaccines", "vaccinology", "vaxx".

When a match occurred, the tweet was analyzed and stored in a database. The analysis involved evaluating the sentiment expressed by the tweet's contents, and following any external links it contained. In the following, we present details regarding the link analysis, the MMR network, and the sentiment classification process.

### Ethics statement

The study, including the data collection process, has received written approval from the Institutional Review Board at the Technical University of Denmark (IRB number COMP-IRB-2021–09).

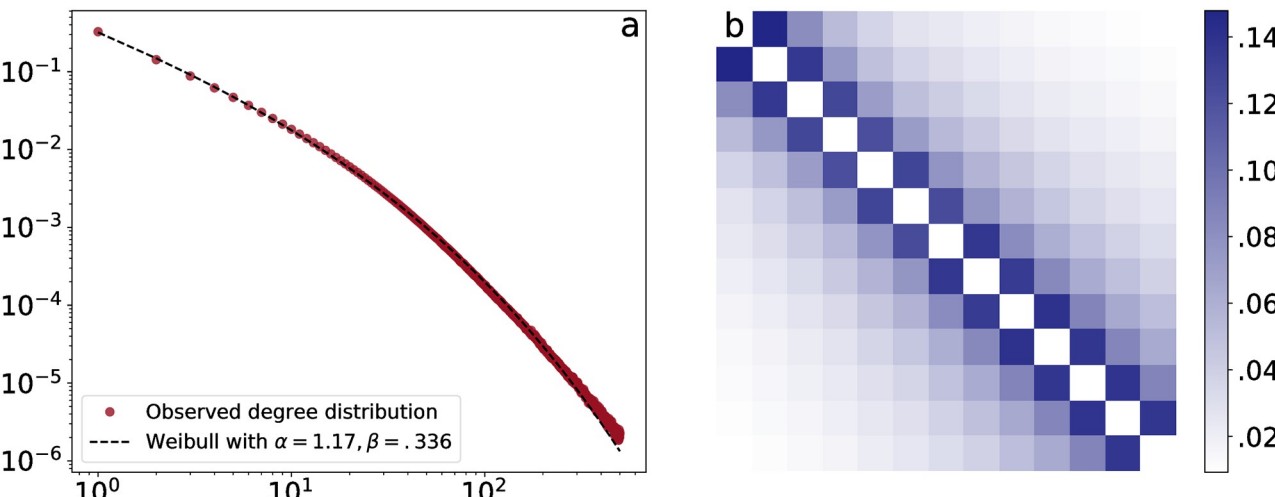

**Fig 7. a**: The degree distribution of the MMR graph, truncated at degree 500 to exclude automated profiles. The dashed line indicates the best fit for a stretched exponential (Weibull) function. **b**: The Jaccard similarity index of the sets of edges in the MMR graph for different 3-month periods. Each row and column correspond to a three-month time windows in the period from 2013–2016. The diagonal is therefor left out, as it represents the self-similarity of the interaction network in each time window, and so the Jaccard similarity is 1 by construction.

## Network analysis

Using the sample of approximately 60 billion tweets (Dataset 1), we construct an interaction network, in which profiles are connected only if both profiles interact with the other, by retweeting content, or by replying, within a three month time window. Each such time window contained an average of 19,588,474 nodes and 47,115,188 edges. Combining the graph for all time windows resulted in a graph with a total of 89,577,277 nodes and 434,193,958 edges. The degree distribution—the probability distribution over $k$, the number of such connections to each profile—for the combined graph is showed in Fig 7a. The degree distribution closely matches a Weibull distribution,

$$p_k = \frac{\beta}{\alpha}\left(\frac{k}{\alpha}\right)^{\beta-1} e^{(k/\alpha)^\beta}, \tag{1}$$

with the parameter values specified in the figure legend.

To assess the stability of the interaction network over time, we constructed an MMR network for each three-month time window in the period 2013–2016. For every pair of such networks, we compared how similar the connections in the two networks were, using the Jaccard similarity index

$$H_{S_i,S_j} = \frac{|S_i \cap S_j|}{|S_i \cup S_j|}. \tag{2}$$

If the Jaccard similarity between the networks at two time points is 1, the network is completely unchanged, and if it is 0, no connection between any two profiles exists at both time points. Fig 7b shows the Jaccard similarities between the MMR networks for all of the three-month time windows. It shows that the similarity is relatively large at neighboring time points, with 14% of connections appearing at both time points, following which the self-similarity over time gradually reduces to almost zero over the period of 3 years.

## Link analysis

Links contained in tweets are shortened by Twitter, and sometimes by external URL shorteners as well. In order to analyze external URLs contained in the vaccine-related tweets, we used python to crawl each URL, repeatedly follow redirects, end noting the domain of the final destination. For profiles that were categorized as strongly anti- or pro-vaccine, we recorded the ten most frequently used such domains. This resulted in a total of 18 domains, due to youtube and facebook occurring in both top tens.

   We manually assigned one or more categories to these 18 domains. The categories, along with classification criteria, are outlined below. For classifying pages as conspiracy or pseudo-science sites, we looked up the domains on the online service *media bias/fact check* (MBFC).

- **Commercial**—Pages that include an online store selling health related products.
  naturalnews.com, articles.mercola.com, go.thetruthaboutvaccines.com, greenmedinfo.com, healthimpactnews.com, healthnutnews.com, infowars.com, newstarget.com, vaccineimpact. com.

- **Conspiracy**—Classified using MBFC.
  awarenessact.com, newspunch.com, newstarget.com, worldtruth.tv, collective-evolution. com, inshapetoday.com, realfarmacy.com.

- **News**—Known news sites.
  bbc.co.uk, bbc.com, bioportfolio.com, cbc.ca, choice.npr.org, cnn.com, edition.cnn.com, forbes.com, foxnews.com, huffingtonpost.com, medicalnewstoday.com, nbcnews.com, nytimes.com, reuters.com, sciencedaily.com, statnews.com, theguardian.com, time.com, whitehouse.gov.

- **Pseudoscience**—Classified using MBFC.
  collective-evolution.com, inshapetoday.com, realfarmacy.com, seattleorganicrestaurants. com.

- **Science**—Sites promoting mainstream science/medical information.
  bioportfolio.com, cdc.gov, medicalnewstoday.com, sciencedaily.com, statnews.com, webmd. com.

- **Social Media**—Large social media platforms.
  facebook.com, instagram.com, reddit.com.

- **Youtube**—The popular video-sharing platform.
  youtube.com.

## Classification of vaccine-sentiment in tweets

Using machine learning techniques in general often requires a large amount of 'ground truth' data on which ones model can be trained, and this is particularly true for deep learning models. Establishing a ground truth dataset often requires human work and is thus often costly. One technique to work around this issue is *transfer learning*, in which a model is first trained on a large 'source' dataset, in which the ground truth has already been established, and then applied to a 'target' dataset. In the case of deep neural networks, this approach typically consists of first training a complex model to the source data, then stripping off the layer of output neurons and using the output of the second last layer, often called a representation layer, in conjunction with another model to predict the desired target dataset. This sometimes increases

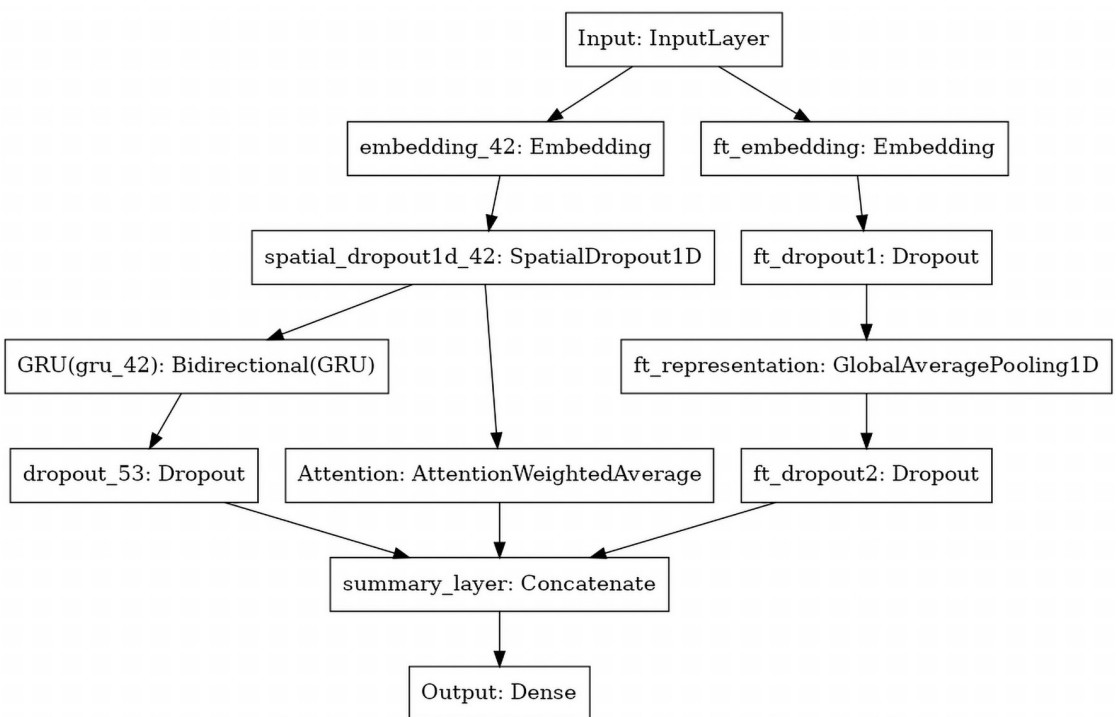

**Fig 8. Representation of the final classifier.** An initial input layer, in which strings are represented by a sequence of one-hot encoded words, is passed to a) a deep neural network similar to the DeepMoji classifier [70], and b) a fasttext classifier [71]. After being pre-trained to predict hashtags from the surrounding text (source dataset), the model is fine-tuned to instead predict vaccine sentiment from tweet text (target dataset).

model performance, as it allows one to 'reuse' higher-order representations of the input data learned by the original classifier. We here describe the source and target datasets, along with the final model architecture, which is summarized in Fig 8.

The **target** data consisted of 10000 randomly selected tweets containing vaccine related keywords. We hired workers on Amazon's Mechanical Turk (MTurk) platform to classify tweets as being either for, or against human vaccination, or as undecideable or unrelated. To ensure high-quality ratings, we first manually rated 100 tweets, then hired a number of MTurk workers for a test assignment which clearly stated that top performers would receive offers for additional tasks. The payment was set to be very high compared to typical MTurk to provide incentive for good performance. We then identified top performers whose scores where most similar to our own, and launched the remaining tasks, allowing only the identified workers to participate. We hired workers such that each tweet would be rated by 3 distinct raters. We then kept only the tweets for which all 3 raters agreed on a label, which reduced the data set to 5358, the distribution of labels in which was 18.8% antivaxx, 45.67% provaxx, and 35.50% neutral/unrelated.

As the **source** dataset, we chose to train the classifier to predict a number of hashtags which we presumed to be related to the sentiment prediction task. From an initial qualitative analysis of the data, and from a brief review of the literature, we noted that

- Anti-vaccine narratives occasionally supposes underlying conspiracies, as represented by hashtags such as #cdctruth, or #cdcwhistleblower.

- Many tweets that that mention vaccine-related keywords are not concerned with vaccination of humans, but rather of pets. To help the classifier disambiguate, we included hashtags such as #dog and #cat.

- There is a relatively popular indie rock band called *The Vaccines*. To help disambiguate, we included hashtags like #music and #livemusic.

Based on the above observations, we opted to scrape for our source dataset a large number of tweets containing any of the following hashtags: #endautismnow, #antivax, #autism, #autismismedical, #cat, #cdctruth, #cdcwhistleblower, #dog, #ebola, #flu, #health, #hearthiswell, #hpv, #immunization, #livemusic, #measles, #medication, #music, #polio, #sb277, #science, #vaccination, #vaccine, #vaccines, #vaccinescauseautism, #vaccineswork, #vaxxed.

Using a large number of tweets ($\approx$ 10,670,000 in total) of tweets containing either of those hashtags, and trained a deep neural network classifier to predict the hashtags from text. These tweets were obtained in a similar fashion to dataset 2. We used a random upsampling approach to achieve a balanced dataset within each training sample when doing cross-validation [70].

The classifier consisted of an embedding layer, a spatial dropout, then a parallel sequence of a) a bi-directional GRU (gated recurrent unit) and a dropout layer, and b) a weighted attention average layer [70]. Those were then concatenated into a representation layer.

After fitting the hashtag model, we removed the output layer and 'froze' the remaining layers, to prohibit training of the weights contained in the original model. We then added a fasttext network [71] in parallel with the pretrained classifier. The rationale for this was that, while the initial classifier might have learned to recognize highly complex patterns in text, it might not do a good job of making simpler connections between input text and target probabilities. After fitting the fasttext part of the classifier, we used the chain-thaw approach of [70] to further improve performance.

On the three-class prediction task, the classifier attained a micro-averaged F1-score of 0.762. The score was computed by aggregating true and false positives/negatives over a 10-fold stratified cross-validation procedure [72]. For comparison with the literature, we also trained the classifier for binary prediction (i.e. predicting simply whether a text snippet was anti-vaxx or not). The accuracy on the binary case was 90.4±1.4% over a 10-fold stratified cross-validation evaluation, an increase over what to our knowledge is state of the art performance [46].

Looking qualitatively at the performance of the classifier, the tweets that were labeled with high confidence demonstrate some capability of the classifier to recognize relatively subtle indications of the correct label for the tweet, as shown in Table 1.

**Table 1. Qualitative summary of classifier performance.** The classifier correctly assigns a large probability of antivaxxness to text snippets the express conspiracist notions about vaccines being part of a global scam. Similarly, texts highlighting the positive qualities of vaccinations are assigned a high probability of being provaxx. In addition, text snippets concerning the band named The Vaccines are recognized as irrelevant. A text snippet expressing how much more expensive it is to kill, rather than vaccinate, badgers is also categorized as irrelevant with a high certainty, despite containing negative words like 'kill'.

| Text | Class |
|---|---|
| "January is Cervical Health Awareness Month.Join the HPV vaccine campaign to prevent cervical cancer" | Provaxx |
| "Getting a flu shot will help prevent transmission to new babies who are too young to be vaccinated. #flukills" | Provaxx |
| "things i love about the vaccines: they change their set list every gig but norgaard is always the last song they play" | Neutral |
| "Hi ⟨USER⟩ £7.3m to kill 1771 Badgers—£4100 per Badger—Vaccination looks cheap now? £662 per Badger?" | Neutral |
| "Bill Gates Admits #Vaccines Are Used for Human #Depopulation" | Antivaxx |
| "Lead Developer Of HPV Vaccines Comes Clean, Warns Parents & Young Girls It's All A Giant Deadly Scam" | Antivaxx |

## Categorization of Twitter profiles from tweets

For each user, we considered tweets containing vaccination-related keywords (see description of Dataset 2 above). For each such tweet, we estimated the probability of the tweet expressing sentiment that is pro-vaccine, anti-vaccine, or neutral/unrelated, using the machine learning classification method described above. We then label profiles as anti/pro-vaxx if the classifier assigns more than 50% of the profile's tweets a probability of at least 50% of being anti/pro-vaxx. Note that this leaves the majority of profiles not assigned to either of the two categories, as illustrated in Fig 1. This strong criterion is intended to reduce the number of profiles falsely assigned into either category.

## Note on uncertainties and robustness

Most of the figures presented here are produced using a very large number of data points. For this reason, some quantities, such as the tweet sentiment and user stance distributions presented in Fig 1, will have very small error bars that are difficult to meaningfully visualize. Meanwhile, the distributions turn out to be more sensitive to changes in the arbitrary threshold for labeling user stances from tweet sentiments, although this does not qualitatively alter the results. In such cases, we have opted to present the figures without error bars in the main paper, referring the reader to S1 Appendix for a more detailed overview of uncertainties, as well as analyses of robustness to the aforementioned sentiment threshold.

## Supporting information

**S1 Appendix. Robustness and uncertainties.** More details on uncertainties and robustness to sentiment thresholds is provided in S1 Appendix.
(PDF)

**S2 Appendix. Geographical analysis of tweets originating from the USA.** A short analysis of tweet sentiment by American state, of potential relevancy to researchers interested in the interplay between state policy/regulations and Twitter discourse, is presented in S2 Appendix.
(PDF)

**S3 Appendix. Temporal evolution of link frequencies.** S3 Appendix presents illustrations of how the number of links posted to each external domain by changes over time.
(PDF)

**S4 Appendix. Explorative analyses of user and neighborhood stances.** S4 Appendix contains some explorative visualizations on the interplay between user stance strength, the strength of disagreement with the user neighborhood, and the neighborhood activity and number of neighbors, for the data underlying Fig 4.
(PDF)

## Acknowledgments

The authors wish to thank Alan Mislove for his invaluable help with collection and analysis of Twitter data, and Bjarke Felbo for sharing his wisdom of machine learning.

## Author Contributions

**Conceptualization:** Bjarke Mønsted, Sune Lehmann.

**Data curation:** Bjarke Mønsted, Sune Lehmann.

**Formal analysis:** Bjarke Mønsted.

**Funding acquisition:** Sune Lehmann.

**Investigation:** Bjarke Mønsted.

**Methodology:** Bjarke Mønsted.

**Project administration:** Bjarke Mønsted, Sune Lehmann.

**Software:** Bjarke Mønsted.

**Supervision:** Bjarke Mønsted, Sune Lehmann.

**Validation:** Bjarke Mønsted.

**Visualization:** Bjarke Mønsted.

**Writing – original draft:** Bjarke Mønsted.

**Writing – review & editing:** Bjarke Mønsted, Sune Lehmann.

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
