## [Decision Letter · Decision Letter 0]

6 Oct 2021

PONE-D-21-27127Characterizing polarization in online vaccine discourse – a large-scale studyPLOS ONE

Dear Dr. Mønsted,

Thank you for submitting your manuscript to PLOS ONE. After careful consideration, we feel that it has merit but does not fully meet PLOS ONE’s publication criteria as it currently stands. Therefore, we invite you to submit a revised version of the manuscript that addresses the points raised during the review process. The reviewers are enthusiastic about the findings and the importance of the work, but offer suggestions to improve the presentation and strengthen the applicability of the results. 

We look forward to receiving your revised manuscript.

Kind regards,

Christopher M. Danforth

Academic Editor

PLOS ONE

Journal Requirements:

2. In your Methods section, please include additional information about your dataset and ensure that you have included a statement specifying whether the collection method complied with the terms and conditions for the website.

Furthermore, you indicated that ethical approval was not necessary for your study. We understand that the framework for ethical oversight requirements for studies of this type may differ depending on the setting and we would appreciate some further clarification regarding your research. Could you please provide further details on why your study is exempt from the need for approval and confirmation from your institutional review board or research ethics committee (e.g., in the form of a letter or email correspondence) that ethics review was not necessary for this study? Please include a copy of the correspondence as an "Other" file.

[The authors wish to thank Alan Mislove for his invaluable help with collection and analysis of Twitter data, and Bjarke Felbo for sharing his wisdom of machine learning. The research described in this paper was funded by the Danish Council for Independent Research, grant number 4184-00556a.]

 [The authors received no specific funding for this work.]

5. We note that Figure 2 in your submission contains map images which may be copyrighted. All PLOS content is published under the Creative Commons Attribution License (CC BY 4.0), which means that the manuscript, images, and Supporting Information files will be freely available online, and any third party is permitted to access, download, copy, distribute, and use these materials in any way, even commercially, with proper attribution. For these reasons, we cannot publish previously copyrighted maps or satellite images created using proprietary data, such as Google software (Google Maps, Street View, and Earth). For more information, see our copyright guidelines: http://journals.plos.org/plosone/s/licenses-and-copyright.

a) You may seek permission from the original copyright holder of Figure 2 to publish the content specifically under the CC BY 4.0 license.  

Reviewers' comments:

Reviewer's Responses to Questions

**Comments to the Author**

1. Is the manuscript technically sound, and do the data support the conclusions?

Reviewer #1: Yes

Reviewer #2: Yes

Reviewer #3: Yes

2. Has the statistical analysis been performed appropriately and rigorously? 

Reviewer #1: No

Reviewer #2: Yes

Reviewer #3: No

3. Have the authors made all data underlying the findings in their manuscript fully available?

Reviewer #1: Yes

Reviewer #2: No

Reviewer #3: Yes

4. Is the manuscript presented in an intelligible fashion and written in standard English?

Reviewer #1: Yes

Reviewer #2: Yes

Reviewer #3: Yes

5. Review Comments to the Author

Reviewer #1: Abstract

• The characterization of all vaccine discussions as against, for, and neutral is an oversimplification. Decades of vaccine hesitancy research has led to the understanding most individuals are not against per say, but “skeptical,” referred to as vaccine hesitancy, or the “middle ground” in your conceptualization. Almost everyone has some type of hesitancy, and hesitancy depends on the vaccine and context they are in; however, the characterization here has equated hesitancy to a rejection stance, which is entirely inappropriate in my view and fully against current vaccine hesitancy discourse.

Introduction/Findings

• I understand the contention for categorizing all profiles and tweets along a dichotomy of against or for vaccines. However, myself and others who know this literature well would find the findings in this paper much much more valuable if the authors were able to develop a more complex characterization that is not dichotomous. As said in the previous comment, the characterization is largely misrepresented. I would suggest authors to clearly outline how they are defining vaccine hesitancy (or related terms) using terms that are widely used by scholars in this area.

Discussion

• I would have liked to see a much more elaborate exploration of the implications of your findings for the literature and policymaking. I get the impression that the wealth of literature on vaccine hesitancy is largely unexplored or ignored. However, this may not be completely accurate. For example, commercial interests have actually been explored in literature in the past few years. These interests are primarily from the focal point of trust, whereas mistrust is directed towards government, providers, and pharma, to direct trust towards companies or groups that do not fall in the canonical communities. These issues have been explored and should be mentioned in your discussion. I would suggest a more comprehensive searching of the literature, including recently published reviews on the topic. Furthermore recent work on COVID-19 vaccine hesitancy have also explored some of these topics, and they are relevant for today’s discussions.

Reviewer #2: I very much enjoyed reading “Characterizing polarization in online vaccine discourse — a large-scale study.” I have only some minor follow-up questions, potential areas for more theoretical expansion, and a handful of suggestions that the authors can take or leave.

Perhaps this is just my taste, but my main suggestion for the authors is to do more to situate their very impressive data collection and methodology and really drive home what they think we’ve learned from all of this. The findings are each individually interesting and do stand on their own, but I think they could be tied together into a more cohesive story. To me, there are at least two fascinating findings here that quantify dynamics that have previously been identified qualitatively — and are each more interesting than the “polarization” referenced in the title (after all, why wouldn’t we expect vaccine discourse to polarize in a venue like Twitter, where people tend to express stances that are more strongly held?).

First, the grift. Journalistic accounts have documented how various conspiracy theorists on the internet try to channel their followers’ fears into product sales, but this is the first time I have seen it quantified. Second, and more importantly, the interaction network. So much work on vaccine misinformation is about exposure to various sources that are flagged as unreliable or misleading — with the implication that if we were better at flagging or removing bad information from these sites, people’s attitudes and behaviors would change in pro-social ways. This work treats Twitter as social *media.* What I take from these findings is that this is not even half of the story — the real action is in the communities that get built on these sites. Which is to say, the really compelling part of this manuscript is its treatment of Twitter as a social *network.* I suspect that no amount of flagging unreliable domains will deter users in these communities from reinforcing each others’ anti-vaccine views — Motta, et al (2021) on anti-vaccine social identities (https://www.tandfonline.com/doi/full/10.1080/21565503.2021.1932528?journalCode=rpgi20) could be a helpful tie-in on this front.

This presents some interesting and tough questions concerning the tradeoffs associated with so much information flowing in these online spaces. Here, Twitter is doing exactly what it’s designed to do: bring people together to share and discuss information. It’s agnostic as to the ends for which it is bringing people together, and it’s legitimately difficult to say with any certainty what we should expect them to do regarding *communities of users* who reinforce each others’ anti-vaccine views. That’s a harder problem than picking some unreliable information sources to flag or even block.

Some follow-up questions:

- How sensitive are these results to higher thresholds for classifying users’ stances? 50% of on-topic tweets with at least 50% p(stance) is an intuitive threshold, but my expectation would be that most tweets with a stance express an unambiguous stance — i.e. p(stance) well over .5 — and most users with a stance are much more consistent than 51%. How much data do we lose when we require more confidence in our classifications?

- Does downsampling instead of upsampling matter? There’s a ton of data here and also a ton of imbalance, so I worry that the relatively smaller number of anti-vax tweets wind up getting overworked.

Remaining suggestions:

- Personally, the geolocation and state-level analysis didn’t do much for me. The share of tweets/users who could be geolocated in the first place is so small, and those users are likely different from users in similar locations who can’t be located. I wouldn’t put my foot down and say it’s wrong or needs to be cut, but I would say that I wouldn’t miss it (it doesn’t even get a full paragraph in the body as it stands). I’m just not sure what it adds.

Reviewer #3: The paper presents an interesting analysis of how Twitter behavior differs based on vaccine stances, in particular there are notable differences in the links that are shared and the degree of connection. I think the stance detection in in individual tweets is interesting methodologically and demonstrated quite rigorously. However, I think the conclusions of the analysis depend on certain assumptions made by the authors and sensitivity to those assumptions is not fully addressed. Moreover, the paper describes differences between two groups (provax and antivax accounts) but does not include uncertainty metrics for the comparisons. Finally, the authors situate their conclusions in the context of broader understandings of vaccination-related issues but not relative to other work studying discussion of vaccinations on Twitter. Acknowledging these connections would substantially improve the paper.

Major issues presented in order of importance.

1. Threshold for profile classification. Nearly all of the analysis depends on grouping accounts into provax, antivax or neutral based on a 50% threshold. While authors note that the 50% threshold is over all tweets not just vaccine-related tweets in the Materials and Methods section (to allow neutral classifications), it is necessary to note if the results are sensitive to this threshold. In particular, any account that tweets frequently about vaccines will be classified as either pro- or antivax with this procedure even if the classification model is uncertain about the actual labels. This discrete labeling from a continuous measure is a fine practice in general, but knowing if the results differ by the cutoff is necessary. In particular, smaller numbers of provax and antivax accounts will affect the uncertainty measures requested in the next point.

2. Uncertainty measures. Throughout the paper, I think the authors do not adequately report uncertainty in their measures of differences between provax and antivax accounts. In particular, the number of accounts labeled as antivax is a small subset of the full data, as shown in Figure 1. Comparing point estimates of distributions, such as in Figures 3, 4 and 7, is likely to show large differences when one group is small just due to random chance. Showing uncertainty with either confidence intervals or p-values, for example, is necessary to demonstrate that the observed differences in point estimates are not likely to have arisen due to random chance.

3. Prior work studying vaccination discourse on Twitter. Discussing prior work on the issue of vaccine debates on Twitter is necessary to present a rigorous analysis, especially to convince a reader that the authors’ modeling choices are reasonable. This paper discusses prior work on the effects of antivax movements on health outcomes, but makes no mention of prior work using Twitter data to study vaccine sentiment. In particular there is a lot of research on echo chambers in vaccination debates, both COVID-19 and non-COVID-19 related. Either identifying other papers that have employed similar methods or explaining why the authors have employed different methods would improve the rigor of the analysis.

4. Correlation construction. Figure 6 shows a correlation between vaccine stance for different kinds of edges provax-provax, antivax-antivax, and provax-antivax. However, those categories are identified by that same vaccination stance. So by construction, provax-provax and antivax-antivax node pairs will be positively correlated and provax-antivax edges will be negatively correlated. I think a global metric, such as correlation across all types of nodes, is an interesting statistic and it would be more useful here. However, using correlation between node features does require identifying an ordering for the nodes or permutations to properly measure the correlation.

Additional analyses. There are a few points in the paper that would benefit from additional analysis.

1. (Major) MMR graph (Figure 5). I think this graph is an excellent visualization of the disconnectedness of provax and antivax accounts. However, I think more should be discussed in the results.

a. Figure 5 only shows the largest connected component. For the parts of the full graph not shown, are they mostly “pure” (all anti- or pro- vax)? Does the connectedness look different for those smaller graphs?

b. What kind of content are the edges formed on? My reading is that the edges indicate interaction between accounts on any kind of tweet, not just vaccine-related ones. Does the anti-vax cluster interact on vaccine related tweets?

c. Expanding on the point above, it would be nice to see if they authors can identify whether the clustering by vaccine stance is attributable to the stance or some other characteristic, i.e. are the antivax accounts densly connected because of their antivax stance or are they connected because they all share some other common interest and happen to be antivax. Establishing this may be challenging due to lack of reliable account-level features on Twitter data, so I understand if the authors are not able to conclusively answer this issue.

d. There are some anti-vax accounts scattered throughout the pro-vax portion of the network and a few pro-vax accounts in the anti-vax cluster. Can you say anything about what makes those accounts (those with many connections to accounts with different stances) different from the clustered accounts? For example, are those people less anti-vax than the clustered portion? A scatterplot of each node’s vaccine stance and proportion of the node’s neighbors who share the same stance would get at this question.

2. (Minor) Time trends in link sharing. The dataset used to analyze link sharing (Dataset 2) spans a large time period. Do certain domains change in popularity overtime? I would think that the popularity of less-reputable sites shared by antivax accounts could ebb and flow over as their reputation changes.

3. (Minor) Geographic distribution. I do not fully see the point if including the geographic distribution of the anti-/provax tweets across US states. Is there something that explains the variation across states, maybe some demographic features of the states? I don’t think this figure is necessary to include if not, especially because so few tweets are geocoded with an identifiable state.

Minor issues.

1. Why use two datasets instead of one? I think there are good reasons for why Dataset 1 is better for conducting network analysis and Dataset 2 is better for analyzing link sharing, and it would be useful for readers to explain that in the paper.

2. What inputs are used in the first layer of the tweet-level classifier? Are they word counts or some word embedding outputs?

3. In Figure 6, I do not see the need to show different curves on the same plot with different y-axes. I think the clarity of the figures would improve if split into four figures rather than two combined ones.

4. Data availability. I understand privacy concerns regarding sharing the text of tweets and account identifiers, but I do think suitably anonymized data can be shared. In particular, Dataset 1 could be shared including only the tweet-level classification scores and indicators for what base URLs were included in the tweets. Dataset 2 could be shared as just a list of interactions and account classification labels.

6. PLOS authors have the option to publish the peer review history of their article (what does this mean?). If published, this will include your full peer review and any attached files.

Reviewer #1: **Yes: **Umair Majid

Reviewer #2: **Yes: **Jon Green

Reviewer #3: No

---

## [Author Response · Author response to Decision Letter 0]

3 Jan 2022

The resubmission contains a rebuttal letter which provides both a high-level summary of the major changes, as well as detailed point by point responses to each individual comment.

---

## [Decision Letter · Decision Letter 1]

26 Jan 2022

Characterizing polarization in online vaccine discourse – a large-scale study

PONE-D-21-27127R1

Dear Dr. Mønsted,

We’re pleased to inform you that your manuscript has been judged scientifically suitable for publication and will be formally accepted for publication once it meets all outstanding technical requirements.

Kind regards,

Christopher M. Danforth

Academic Editor

PLOS ONE

Additional Editor Comments (optional):

Reviewers' comments:

Reviewer's Responses to Questions

**Comments to the Author**

1. If the authors have adequately addressed your comments raised in a previous round of review and you feel that this manuscript is now acceptable for publication, you may indicate that here to bypass the “Comments to the Author” section, enter your conflict of interest statement in the “Confidential to Editor” section, and submit your "Accept" recommendation.

Reviewer #2: All comments have been addressed

Reviewer #3: All comments have been addressed

2. Is the manuscript technically sound, and do the data support the conclusions?

Reviewer #2: Yes

Reviewer #3: Yes

3. Has the statistical analysis been performed appropriately and rigorously? 

Reviewer #2: Yes

Reviewer #3: Yes

4. Have the authors made all data underlying the findings in their manuscript fully available?

Reviewer #2: Yes

Reviewer #3: Yes

5. Is the manuscript presented in an intelligible fashion and written in standard English?

Reviewer #2: Yes

Reviewer #3: Yes

6. Review Comments to the Author

Reviewer #2: I was enthusiastic about this manuscript on the first round, and the authors have done an excellent job of addressing my minor concerns and incorporating my feedback. As far as I'm concerned, I look forward to seeing it published!

Reviewer #3: (No Response)

7. PLOS authors have the option to publish the peer review history of their article (what does this mean?). If published, this will include your full peer review and any attached files.

Reviewer #2: **Yes: **Jon Green

Reviewer #3: No

---

## [Editor Report · Acceptance letter]

31 Jan 2022

PONE-D-21-27127R1 

Characterizing polarization in online vaccine discourse - a large-scale study 

Dear Dr. Mønsted:

I'm pleased to inform you that your manuscript has been deemed suitable for publication in PLOS ONE. Congratulations! Your manuscript is now with our production department. 

Kind regards, 

on behalf of

Dr. Christopher M. Danforth 

Academic Editor

PLOS ONE